# Green Extraction-Assisted Pseudo-Targeted Profile of Alkaloids in Lotus Seed Epicarp Based on UPLC-QTOF MS with IDA

**DOI:** 10.3390/foods11071056

**Published:** 2022-04-06

**Authors:** Xiaoji Cao, Xupin Lin, Congcong Wu, Minghua Zhang, Mingwei Wang

**Affiliations:** 1Research Center of Analysis and Measurement, Zhejiang University of Technology, Hangzhou 310014, China; 2College of Chemical Engineering, Zhejiang University of Technology, Hangzhou 310014, China; xupinl@163.com (X.L.); claire1630@163.com (C.W.); minghuaz2022@163.com (M.Z.); mingwei96149@163.com (M.W.)

**Keywords:** lotus seed epicarp, alkaloids, bio-based solvents, pseudo-targeted profile, UPLC-QTOF MS

## Abstract

Lotus seed epicarp, a byproduct of lotus, is commonly discarded directly or burned in the cropland, resulting in waste of resources and environmental pollution. In this work, a green ultrasonic-assisted extraction method with ethyl lactate as the extraction solvent was established to extract alkaloids from lotus seed epicarp. The extraction conditions were optimized by response surface methodology. Under the optimal extraction conditions, the extraction of alkaloids from 1 g lotus seed epicarp was accomplished with only 10 mL of extraction solvent within 15 min. Combined with ultrahigh-performance liquid chromatography coupled with quadrupole time-of-flight tandem mass spectrometry with information-dependent acquisition mode, a total of 42 alkaloids were annotated in the lotus seed epicarp extracts. Among them, 39 alkaloids were reported in lotus seed epicarp for the first time. According to quantitative analysis, the distributions and trends of alkaloids in the lotus seed epicarp were found to be similar to those of lotus leaves. The five growth stages of lotus seed epicarp could be successfully distinguished based on the ten representative alkaloids. This study demonstrates that ultrasonic-assisted extraction with ethyl lactate as extractant solvent was efficient in the extraction of alkaloids from lotus seed epicarp, which is a potential renewable resource of bioactive ingredients.

## 1. Introduction

Lotus, a plant of perennial aquatic crop, is widely cultivated in Asia and Oceania. The lotus grown in China mainly belongs to the family of *Nelumbo nucifera* Gaertn., which is widely cultivated in Hubei, Jiangsu, Anhui, Zhejiang and Jiangxi provinces [1]. Lotus is well known as a homology of medicine and food; lotus seeds and lotus roots are used as food in daily life. Other parts of lotus, such as leaves and plumules, have been reported to contain a variety of alkaloids, with pharmacological activities treating hyperlipidemia and adiposity, in addition to anticancer, antioxidation liver-protecting activities [2,3,4]. Lotus seed epicarp is a byproduct produced during lotus seed processing [5,6]; it is almost always discarded directly, leading to a waste of resources and environmental pollution. At present, most research is focused on flavonoids, polyphenols and proanthocyanidins of lotus seed epicarp [7,8,9,10], but there are few reports about the alkaloids of lotus seed epicarp.

The alkaloids in lotus have been successfully extracted by reflux extraction using methanol, ethanol or trichloromethane as the extraction solvent [11,12,13,14], which has many disadvantages, such as the low extraction efficiency and the high consumption of solvent [15]. Ultrasound-assisted extraction has been widely applied in the extraction of bioactive compounds from plants due to its characteristics of less solvent and energy consumption, simple operation and good reproducibility [16,17,18,19]. Ultrasonic-assisted extraction destroys the cell walls of plants through an acoustic cavitation effect produced by the ultrasonic wave and accelerates the release of bioactive compounds in cells to solvents to improve the extraction efficiency [20]. However, traditional organic solvents with high vapor pressure are easy to volatilize into the air during the process of ultrasonic-assisted extraction, which would decrease the extraction efficiency [21]. Therefore, it is necessary to find green and efficient solvents to replace traditional organic solvents.

Bio-based solvents such as glycerol, D-limonene and ethyl lactate are considered green solvents, derived from renewable sources such as corn, wheat or residual organic matter considered waste [22]. Ethyl lactate has the advantage of low vapor pressure and is non-toxic, non-corrosive, biodegradable, renewable and approved by the U.S. Food and Drug Administration (USFDA) as a pharmaceutical and food additive [23]. Moreover, it is soluble both in polar and non-polar solvents. As an alternative to traditional solvents, ethyl lactate has been successfully applied to extract bioactive compounds, such as carotenoids, polyphenols and caffeine, from various plants due to its high extraction efficiency and excellent solubility properties [24,25,26,27]. However, the application of ethyl lactate as the extraction agent to obtain alkaloids from lotus seed epicarp has not been reported until now.

Ultra-high-performance liquid chromatography coupled with quadrupole time-of-flight tandem mass spectrometry (UPLC-QTOF-MS/MS) with information-dependent acquisition (IDA) has the ability to obtain precursor ions and fragment ions simultaneously with high accuracy, which obviously improves the acquisition speed for identification analysis. Therefore, it has been widely applied in the analysis of non-targeted constituents in natural products and metabolites [28,29,30]. 

In this work, an ultrasonic-assisted extraction with ethyl lactate as the solvent was applied to extract alkaloids from lotus seed epicarp. The extraction conditions were optimized by the response surface methodology (RSM). Then, a pseudo-targeted method was developed to profile the alkaloids in lotus seed epicarp by UPLC-QTOF-MS/MS with IDA. Furthermore, the distribution difference of ten alkaloids in lotus seed epicarp at different growth stages was investigated by principal component analysis (PCA). The alkaloids in lotus seed epicarp were systematically studied to provide a reference for further study on chemical components and nutritive value.

## 2. Materials and Methods

### 2.1. Analytical Reagents and Chemicals

HPLC-grade acetonitrile and methanol were purchased from Merck (Darmstadt, Germany), and HPLC-grade formic acid was obtained from Aladdin Reagent (Shanghai, China). Ethanol and ethyl lactate of analytical grade were purchased from Sinopharm Chemical Reagent Co., Ltd. (Shanghai, China).

Standards of alkaloids: nicotinamide (≥98%) and nuciferine (≥98%) were purchased from Aladdin Reagent (Shanghai, China). Asimilobine (≥98%) was purchased from Hubei Qifei Pharmaceutical and Chemical Co., Ltd. (Tianmen, China). *N*-nornuciferine (≥98%), *N*-methylcoclaurine (≥95%) and coclaurine (≥98%) were purchased from Shanghai Yuanye Bio-Technology Co., Ltd. (Shanghai, China). A mixed standard of cis-*N*-feruloyltyramine (≥80%) and trans-*N*-feruloyltyramine (≥18%) was purchased from Shanghai Chengshao Biological Technology Co., Ltd. (Shanghai, China). Isoliensinine and neferine were isolated from Nelumbo nucifera Gaertn. extracts according to the experimental method reported in [14].

### 2.2. Sample Preparation

Lotus seeds were collected from June 2021 to August 2021 in Jinhua (Zhejiang, China). The lotus seed epicarp was manually peeled from the fresh lotus seeds and lyophilized immediately. Subsequently, the lyophilized lotus seed epicarp was ground using a pulverizer (Taisite, Tianjin, China), passed through a 100-mesh sieve and stored at −18 °C prior to extraction.

### 2.3. Ultrasonic-Assisted Extraction

Ultrasonic-assisted extraction was carried out according to the method described in our previous work, with a slight modification [31]. A KQ-100DB ultrasonic cleaning bath (Shumei, Kunshan, Jiangsu, China) with an adjustable ultrasonic power in the range of 10–100 W and a frequency of 40 kHz was employed to extract alkaloids from lotus seed epicarp. A total of 1.0 g lotus seed epicarp powder and 10 mL ethyl lactate aqueous solution (55%, *v*/*v*) was displaced in a 50 mL glass vial and extracted under ultrasound for 15 min, with an ultrasonic temperature of 50 °C and ultrasonic power of 60 W. The extracting solution was centrifuged at 12,000 rpm for 5 min at 4 °C (Hettic, MIKRO 185, Tuttlingen, Germany). Then, 500 µL supernatant was evaporated to dryness by a speed vacuum concentrator (Thermo Fisher Scientific, Waltham, MA, USA). Finally, the residual was reconstituted in 200 µL acetonitrile/water of 2:8 (*v*/*v*) for UPLC-QTOF-MS/MS analysis. 

### 2.4. Optimization of Ultrasonic Extraction Method

#### 2.4.1. Single-Factor Experiments

Single-factor experiments were designed to evaluate the effects of ethyl lactate concentration, solid-to-liquid ratio, ultrasonic time, ultrasonic temperature and ultrasonic power on the total alkaloid content extracted from lotus seed epicarp. The experiments are detailed in Appendix A.

#### 2.4.2. Response Surface Methodology (RSM) Experiments

According to the single-factor experiments, the ethyl lactate concentration, solid-to-liquid and ultrasonic time were selected as three factors to investigate their interaction with the total alkaloid content extracted from lotus seed epicarp using the RSM experiment with a three-factor, three-level Box–Behnken design (BBD). The detailed factors and levels of variables are shown in Appendix A. A total of 17 experimental runs were conducted for the RSM. Statistical analysis was carried out with Design-Expert software (Version 8.0.6.1).

The total alkaloid content is expressed as mg of nuciferine equivalent per gram of dried lotus seed epicarp (mg nuciferine/g). The preparation of nuciferine standard solution and the standard curve (Appendix A) are detailed in Appendix A.

### 2.5. Conventional Reference Extraction Methods

Four extraction methods, including ultrasonic-assisted extraction with ethanol, ultrasonic-assisted extraction with methanol, reflux with ethanol and reflux with methanol, were selected as the reference methods for extraction of the alkaloids from lotus seed epicarp. The experiments are detailed in Appendix A.

### 2.6. Antioxidant Activities

The antioxidant activities of the alkaloids in lotus seed epicarp were evaluated using a DPPH (1,1-diphenyl-2-picryl-hydrazyl) radical scavenging assay, following the previously described methods [6]. Briefly, 6.25 mg DPPH was weighed and dissolved in methanol to 25 mg/L. Volumes 1 µL of sample solutions were incubated with 25 mg/L DPPH in 5 mL methanol solvent at room temperature for 30 min in darkness, and then absorbance at 519 nm was determined by UV-vis spectrophotometer (UV-2600, Shimadzu, Kyoto, Japan). The percentage inhibition of the DPPH radical was calculated as DPPH radical scavenging rate, (%) = (A_1_ − A_2_)/A_1_ × 100, where A_1_ = absorbance of 25 mg/L DPPH alone in methanol, and A_2_ = absorbance of 25 mg/L DPPH with sample in methanol.

### 2.7. Conditions of UPLC-QTOF-MS/MS

An LC-30AD (Shimadzu, Kyoto, Japan) equipped with a Waters ACQUITY Premier HSS T3 column (2.1 mm × 100 mm, 1.8 µm) and Waters Van Guard FIT cartridge (2.1 mm × 5 mm, 1.8 µm) was used for separation. The column temperature was maintained at 35 °C, and the flow rate was 0.4 mL/min. The injection volume was 2 µL. The mobile phase consisted of 0.1% formic acid in water (A) and 0.1% formic acid in acetonitrile (B). The gradient started with 12% B, was held for 3 min, then linearly increased to 65% B within 7 min, held for 1 min, and returned to 12% B for 0.1 min, followed by equilibration with 12% B for 1.9 min.

An AB SCIEX Triple TOF 5600^+^ (AB SCIEX, Foster City, CA, USA) equipped with a DuoSpray ion source was used to obtain the qualitative mass spectral data with information-dependent acquisition (IDA). The ESI ion source was operated in positive ion mode. The voltage of spray was set at 5500 V, and the temperature of the electrospray ion source was 500 °C. Ion source gas 1 (air), ion source gas 2 (air) and curtain gas (N_2_) were set at 55, 55 and 35 psi, respectively. The mass ranges of the full scan and the product ion scan were set at *m*/*z* 100–800 Da and *m*/*z* 50–800 Da, respectively. Dynamic background subtract and the five most intense precursor ions in a cycle of full scan were used to acquire the IDA data. The collision energy (CE) and collision energy spread (CES) were set at 45 V and 15 V, respectively.

An AB SCIEX X500R (AB SCIEX, Foster City, CA, USA) equipped with a Turbo V^TM^ ion source was used to acquire the quantitative data of ten representative alkaloids with high-resolution multiple-reaction monitoring (MRM^HR^). The ESI ion source was operated in positive ion mode. The voltage of spray was set at 5500 V, and the temperature of the electrospray ion source was 500 °C. Ion source gas 1 (air), ion source gas 2 (air) and curtain gas (N_2_) were set at 55, 55 and 35 psi, respectively. The parameters of MRM^HR^ experiments for 10 representative alkaloids were set as reported in Appendix A. 

All data acquisition and analysis were performed with SCIEX OS (Version 2.1.6) analysis software (AB SCIEX, Foster City, CA, USA).

### 2.8. Quantification of Ten Representative Alkaloids in Lotus Seed Epicarp at Different Growth Stages

To evaluate the performance of the quantitative method by UPLC-QTOF-MS/MS in (MRM^HR^) mode, linearity, LOQs, LODs, accuracy and precision were assessed under the optimized experimental conditions. Linearity was assessed by analyzing standard solutions at six concentrations (5–200 µg/L for nicotinamide and isoliensinine; 1–200 µg/L for *N*-nornuciferine, coclaurine, *N*-methycoclaurine, nuciferine, asimilobine, neferine, cis-*N*-feruloyltyramine and trans-*N*-feruloyltyramine) prepared in methanol. The LODs and LOQs of the method were estimated with S/N ratios of 3 and 10, respectively. Accuracy was evaluated in terms of recovery at three spiked levels (5, 50 and 100 ng/L). Precision was measured in terms of intraday repeatability and interday repeatability.

Totals of 1.0 g lotus seed epicarp powder at different growth stages and 10 mL ethyl lactate aqueous solution (55%, *v*/*v*) were displaced in a 50 mL glass vial and extracted under the optimal extraction method. The extracting solution was centrifuged at 12,000 rpm for 5 min at 4 °C. Then, 500 µL supernatant was evaporated to dryness by speed vacuum concentrator. Finally, the residual was reconstituted in 200 µL acetonitrile/water of 2:8 (*v*/*v*) for UPLC-QTOF-MS/MS quantification analysis.

### 2.9. Statistical Analysis

Principal component analysis (PCA) was carried out by MetaboAnalyst software http://www.metaboanalyst.ca (accessed on 1 March 2022) at 95% confidence level. The concentration of ten representative alkaloids in lotus seed epicarp at different growth stages were acquired with the proposed quantitative method in triplicate, and the results are expressed as mean ± SD (standard deviation). 

## 3. Results and Discussion

### 3.1. Optimization of Extraction Conditions

#### 3.1.1. Single-Factor Experiments

Single-factor experiments were designed to evaluate the effects of ethyl lactate concentration, solid-to-liquid ratio, ultrasonic time, ultrasonic temperature and ultrasonic power on the total alkaloid content extracted from lotus seed epicarp.

As shown in Figure 1A, when the ethyl lactate concentration ranged from 10% to 50% (*v*/*v*), the total alkaloid content increased with increased ethyl lactate concentration. When the ethyl lactate concentration was more than 50%, the total alkaloid content declined dramatically. It is apparent that high viscosity of the extract solution is unfavorable for the mass transfer of alkaloids from lotus seed epicarp to the solution [23]. A reduction in water content in the solution may affect the ability the lotus seed epicarp tissues to swell, impeding penetration of the solvents [26]. Therefore, an ethyl lactate concentration of 50% was selected for subsequent experiments. 

Solid-to-liquid ratio is also an important factor affecting the total alkaloid content extracted from lotus seed epicarp. As shown in Figure 1B, the total alkaloid content gradually increased when the solid-to-liquid ratio ranged from 1:5 to 1:20 g/mL. No significant change was observed when the solid-to-liquid ratio was below 1:20 g/mL, which demonstrates that more solvent could promote the diffusion of alkaloids into the medium until the mass transfer process reached equilibrium [17]. Considering the saving of reagents, a solid-to-liquid ratio of 1:20 g/mL was adopted in the following experiments.

As shown in Figure 1C, the maximum of the total alkaloid content was obtained with 10 min ultrasonication, which indicates that the concentration of alkaloids reached equilibrium between the medium and lotus seed epicarp at 10 min [32]. However, the alkaloids in the extraction would be decomposed with a long extraction time [33]. Thus, an ultrasonic extraction time of 10 min was selected.

As shown in Figure 1D, the ultrasonic temperature had nearly no effect on the extraction of alkaloids from lotus seed epicarp, and an ultrasonic temperature of 50 °C was selected for subsequent experiments.

As shown in Figure 1E, the total alkaloid content reached its maximum value when ultrasonic power was 60 W, which supplied enough energy to destroy the cell wall and facilitated the mass transfer of the alkaloids to the medium. When the ultrasonic power was higher than 60 W, the structure of the alkaloids may have been damaged [16]. Consequently, an ultrasonic power of 60 W was selected.

#### 3.1.2. Response Surface Optimization

Based on the results of single-factor experiments, three independent factors ((A) ethyl lactate concentration, (B) solid-to-liquid ratio and (C) ultrasound time) were chosen to evaluate the impact of the variables on the total alkaloid content extracted from lotus seed epicarp. A total 17 experimental groups were conducted for the response surface methodology (RSM), as shown in Appendix A. 

Analysis of variance (ANOVA) for the response quadratic model is illustrated in Appendix A. The *p*-value of the model was <0.0001, which demonstrated that the model was extremely significant. The F-value of the lack of fit was 0.91, which implies the lack of fit was not significant relative to the pure error. In addition, the value of the determination coefficient (R^2^) and the value of the adjusted determination coefficient (R^2^_Adj_) were 0.9866 and 0.9739, respectively, indicating the model had a satisfactory fitting degree relative to the experimental results. 

The interaction between variables can be intuitively reflected by the 3D response surface. As shown in Figure 2A,B, the response surface plots are obviously curved, which indicates that the interaction effect between ethyl lactate concentration and the solid-to-liquid ratio (P_AB_ = 0.0089), as well as between ethyl lactate concentration and ultrasonic time (P_AC_ = 0.0059) on the total alkaloid content extracted from lotus seed epicarp was statistically significant. However, the interaction effect between the solid-to-liquid ratio and ultrasonic time (P_BC_ = 0.5465) was not significant (Figure 2C). 

According to the results of RSM, the optimal extraction conditions of alkaloids are as follows: ethyl lactate concentration, 55%; solid-to-liquid ratio, 1:10 g/mL; ultrasonic time, 15 min. Under the optimal conditions, the total alkaloid content extracted from lotus seed epicarp was 34.75 ± 1.55 mg nuciferine equivalent/g, which correlates well with the predicted value (the error was only 0.22%). The results show that the proposed method is feasible for optimizing the extraction conditions of alkaloids from lotus seed epicarp. 

#### 3.1.3. Comparison with Previously Reported Methods

The proposed extraction method was compared with previously reported methods. The total alkaloid contents obtained from lotus seed epicarp by different extraction methods are shown in Table 1. It can be seen that the total alkaloid contents obtained from lotus seed epicarp by the ultrasonic-assisted extraction method with ethyl lactate approximated those obtained by the ultrasonic-assisted extraction method with methanol and the reflux extraction method with ethanol. The developed method has the advantage of less organic solvent consumption and shorter extraction time; the extraction of alkaloids from 1 g lotus seed epicarp was accomplished with only 10 mL of extraction solvent within 15 min. It is obvious that ethyl lactate can be considered a green and effective substitute for traditional organic solvents for extraction of alkaloids from lotus seed epicarp.

#### 3.1.4. Antioxidant Activity of Lotus Seed Epicarp Extracts

As shown in Figure 3, the ethyl lactate extract of mature lotus seed epicarp (stage V) displayed the best DPPH radical scavenging rate (79.84%) among the five extracts obtained by different extraction methods.

### 3.2. Pseudo-Targeted Profile of Alkaloids in Lotus Seed Epicarp

Until now, more than 100 kinds of alkaloids have been found in various parts of lotus, such as leaves, seedpods and Plumula nelumbinis [11,37]. However, there have been few investigations of alkaloids in lotus seed epicarp. As a part of lotus, the seed epicarp may have similar chemical constituents to those of other parts. Thus, a pseudo-targeted method for profiling the alkaloids in lotus seed epicarp was developed. According to the literature [3,34,38], the alkaloids in lotus can be divided into four groups, including monobenzylisoquinoline alkaloids, bisbenzylisoquinoline alkaloids, aporphine alkaloids and amide alkaloids. A pseudo-targeted dataset including information on molecular structures, protonated ions, characteristics of fragment ions and the neutral losses of the four types of alkaloids was developed by summarizing all alkaloids of lotus previously reported in the literature. The retention time (t_R_), protonated ions (MS^1^) and fragment ions (MS^2^) of alkaloids in lotus seed epicarp extracts were acquired by UPLC-QTOF MS with IDA mode. Then, the MS^1^ and MS^2^ spectral features of the alkaloids were extracted from the IDA data based on the pseudo-targeted dataset [3,11,34,35,38,39,40,41,42,43,44,45,46]. Subsequently, the alkaloid molecular structures were deduced by analysis of the MS^1^ and MS^2^ spectral data. Moreover, ten alkaloid standards were used to further verify the alkaloids in lotus seed epicarp. A total of 42 alkaloids were tentatively authenticated, including 20 benzylisoquinoline alkaloids, 2 bisbenzylisoquinoline alkaloids, 15 aporphine alkaloids and 5 amide alkaloids. The extracted ion chromatograms (XIC) and the structures of 42 alkaloids proposed in lotus seed epicarp are shown in Figure 4 and Figure 5, respectively. The mass spectral data are detailed in Appendix A.

#### 3.2.1. Identification of Monobenzylisoquinoline Alkaloids

Monobenzylisoquinoline alkaloids are a class of alkaloids with the isoquinoline parent nucleus superseded by a benzyl group at C-1 (Figure 5A). In positive ion mode, these alkaloids tended to form protonated ions because the proton was easily bound at N-2 of the isoquinoline. The most significant feature of these alkaloids is the presence of benzyl ions (*m*/*z* 107.0491, R_4_ = H; *m*/*z* 121.0648, R_4_ = CH_3_) in the MS^2^ spectra. Moreover, if a methyl group or two methyl groups are located at N-2, the neutral loss of a CH_3_NH_2_ (−31.0416 Da) or an NH(CH_3_)_2_ (−45.0573 Da) would be observed in their MS^2^ spectra, respectively. The loss of a NH_3_ (−17.0260 Da) in the MS^2^ spectrum indicates that there was no substituent at N-2. As for the positions of C-6 and C-7, when they are simultaneously superseded by hydroxyl groups, the loss of an H_2_O (−18.0100 Da) would be observed in their MS^2^ spectra. When they are simultaneously replaced by methoxyl groups, the successively losses of two methyl groups (−15.0229 Da) is be observed. When they are replaced by a hydroxyl group or a methoxyl group, respectively, the neutral loss of a CH_3_OH (−32.0257 Da) is observed. In addition, the positions C-7 and C-4′ are often substituted by a hydroxyl group, which is easily glycosylated [11]. The neutral loss of a glucosyl (−162.0521 Da) group could predict the presence of a glucosyl group at C-7 or C-4′. According to the pseudo-targeted dataset, twenty monobenzylisoquinoline alkaloids were profiled from the ethyl lactate extract of lotus seed epicarp.

Compounds **10**, **16**, **20** and **22** have diagnostic ions at [M + H − 17]^+^ in their MS^2^ spectrum, which resulted from the neutral loss of an NH_3_ from their protonated ions. As an example, compound **16** was eluted from an LC column at 4.06 min with a protonated ion ([M + H]^+^) at *m*/*z* 286.1438 (Figure 6). The diagnostic fragment ion at *m*/*z* 269.1179 was generated by the neutral loss of an NH_3_ from the protonated ion. The fragment ions at *m*/*z* 254.0943 and 237.0910 were produced by the loss of a methyl group and a CH_3_OH from the fragment ions at *m*/*z* 269.1179, respectively. In addition, the benzyl ion at *m*/*z* 107.0485 was observed in the MS^2^ spectrum. The spectral data of compound **16** were consisted with those of coclaurine in the developed pseudo-targeted dataset. Comparing the t_R_, MS^1^ and MS^2^ with those of coclaurine standard, compound **16** was identified as coclaurine. By comparison with the pseudo-targeted dataset, compounds **10**, **20** and **22** with protonated ions at *m*/*z* 272.1282, 300.1594 and 286.1438 were preliminarily identified as norcoclaurine, N-norarmepavine and isococlaurine, respectively.

Six alkaloids (compounds **11**, **15**, **17**, **19**, **23** and **28**) have diagnostic ions at [M + H − 31]^+^ in their MS^2^ spectra, which were formed by the neutral loss a NH_2_CH_3_ from their protonated ions. Compound **17**, with a protonated ion at *m*/*z* 300.1592, was easily identified as *N*-methylcoclaurine by comparison of its retention time and the MS^2^ fragment ions with those of *N*-methylcoclaurine in the pseudo-targeted dataset. By comparison with the developed dataset, compound **11**, with a protonated ion at *m*/*z* 286.1438, and compound **19**, with a protonated ion at *m*/*z* 300.1594, were tentatively recognized as *N*-methylnorcoclaurine and armepavine, respectively. Compounds **15** and **23** were tentatively identified as *N*-methylisococlaurine and 6-demethyl-4′-methyl-*N*-methylcoclaurine, respectively, with the same protonated ions at *m*/*z* 300.1594. Moreover, compound **28** had a protonated ion at *m*/*z* 314.1751 and a fragment ion at *m*/*z* 121.0651 in the MS^2^ spectrum, indicating that the C-4′ was substituted with a methoxy group. Combined with the pseudo-targeted dataset, it was temporarily recognized as 4′-methyl-*N*-methylcoclaurine. 

Characteristic ions at [M + H − 162]^+^ were observed in the MS^2^ spectra of nine compounds (**2**, **3**, **4**, **5**, **6**, **7**, **8**, **12** and **14**), which were originated from the loss of a glucose group. By comparison with the pseudo-targeted dataset, Compound **2**, with a protonated ion at *m*/*z* 434.1809, was preliminarily identified as norcoclaurine-6-*O*-glucoside. Compounds **3**, **4**, **6** and **7** produced protonated ions at *m*/*z* 448.1965 and were tentatively recognized as *N*-methylnorcoclaurine 7-*O*-glucoside, N-methylnorcoclaurine 4′-*O*-gluside, coclaurine 7-*O*-glucoside and coclaurine 4′-*O*-gluside, respectively, based on the pseudo-targeted dataset. Similarly, compounds **5** and **8**, with the same protonated ions at *m*/*z* 462.2122, were tentatively assigned as *N*-methylcoclaurine 7-*O*-gluside and *N*-methylcoclaurine 4′-*O*-glucoside. Compounds **12** and **14**, at *m*/*z* 476.2278 and with an elemental composition of C_25_H_33_NO_8_, were tentatively identified as lotusine 7-*O*-glucoside and lotusine 4′-*O*-glucoside, respectively.

Compound **13** has an elemental composition of C_19_H_23_NO_3_, and its mass spectral data were consistent with those of lotusine in the pseudo-targeted dataset. Thus, compound **13** was tentatively regarded as lotusine.

#### 3.2.2. Identification of Bisbenzylisoquinoline Alkaloids

Bisbenzylisoquinoline alkaloids are a class of alkaloids consisting of two monobenzylisoquinoline alkaloids linked by an ether bond (Figure 5B). In positive ion mode, these alkaloids usually formed protonated ions in their MS spectra. If the position of N-2 was superseded by a methyl group, the neutral loss of a CH_3_NH_2_ (−31.0416 Da) could be observed in the MS^2^ spectra. In addition, the cleavage of the bonds between C-1′ and C-9′ and between C-1 and C-9 led to the formation of characteristic isoquinoline ions and benzyl ions, respectively.

Compound **25** was eluted from the LC column at 5.39 min with a protonated ion ([M + H]^+^) at *m*/*z* 625.3276 (Figure 7). The diagnostic fragment ions at *m*/*z* 594.2822 and 582.2856 were produced by the neutral loss of a CH_3_NH_2_ and a CH_3_N=CH_2_, respectively. In addition, the characteristic fragment ions at *m*/*z* 206.1175 and 121.0630 corresponding to the isoquinolin and benzyl ions, respectively, were observed in the MS^2^ spectrum. By comparison of its t_R_, MS^1^ and MS^2^ with those of neferine standard, compound **25** was identified as neferine. Compound **21**, with protonated ions at *m*/*z* 611.3120, was identified as isoliensinine by the same method.

#### 3.2.3. Identification of Aporphine Alkaloids

When the C-8 atom and the C-6′ atom of monobenzylisoquinoline were conjugated, aporphine alkaloids were formed (Figure 5C). In positive ion mode, protonated ions were obviously observed in the MS spectra. The most significant MS/MS spectroscopic feature of these alkaloids was the fragment ion at *m*/*z* 191.0854 [45]. For the majority of aporphine alkaloids, the substitutions of N-2, C-5 and C-6 were the same as those of monobenzylisoquinoline alkaloids. Thus, the loss of a NH_3_ (−17.0260 Da), an NH_2_CH_3_ (−31.0416 Da), an H_2_O (−18.0100 Da), a CH_3_OH (−32.0257 Da) and a methyl group (−15.0229 Da) would be observed in their MS^2^ spectra. When a methylene dioxygen bridge structure was formed between C-6 and C-7, the neutral loss of an HCHO (−30.0100 Da) could be observed [47]. 

Compounds **24**, **29**, **32** and **33** have characteristic ions at [M + H − 17]^+^ in their MS^2^ spectra, which were formed by the loss of an NH_3_ from their protonated ions. By comparison of the retention time and the mass spectral data with standards, compound **24**, with a protonated ion at *m*/*z* 268.1332, and compound **33**, with a protonated ion at *m*/*z* 282.1489, were identified as asimilobine and *N*-nornucififerine, respectively. Compound **29** has a protonated ion at *m*/*z* 268.1332 and fragment ions at *m*/*z* 251.1074, 236.0840, 219.0809 and 191.0856, which are the same as the molecular feature of compound **24**. According to the developed dataset, compound **29** was tentatively identified as caaverine, an isomer of asimilobine. Compound **32** has a protonated ion at *m*/*z* 266.1175 (C_17_H_16_NO_2_^+^). In the MS^2^ spectrum, the characteristic ion at *m*/*z* 249.0910 was formed by the loss of an NH_3_ from the protonated ion. The fragment ion at *m*/*z* 219.0804 was produced by the loss of an HCHO from the fragment ion at *m*/*z* 249.0910. By comparison with the pseudo-targeted dataset, compound **32** was temporarily assigned as anonaine.

The diagnostic ions at [M + H − 31]^+^ were observed in the MS^2^ spectra of four compounds (**26**, **31**, **34**, and **35**), which were formed by the neutral loss of an NH_2_CH_3_ from their protonated ions. Compound **35** was eluted from an LC column at 6.05 min, with protonated ion at *m*/*z* 296.1649 (Figure 8). The fragment ion at *m*/*z* 265.1218 resulted from the neutral loss of a CH_3_NH_2_. The fragment ions at *m*/*z* 250.0985 and 235.0759 were yielded by the successive losses of two methyl groups from the fragment ions at *m*/*z* 265.1218. Moreover, diagnostic fragment ions at *m*/*z* 191.0855 were observed in the MS^2^ spectrum. Thus, compound **35** was identified as nuciferine, which has the same retention time and mass spectral data as those of nuciferine standard. Compounds **26** and **31**, with the same protonated ions at *m*/*z* 282.1489, were tentatively recognized as O-nornuciferine and lirinidine, respectively, by comparison with the pseudo-targeted dataset. Compound **34** has an [M + H]^+^ ion at *m*/*z* 280.1332, and the fragment ions of it at *m*/*z* 249.0911, 219.0801, 191.0854 in the MS^2^ spectrum are the same as those of romerine in the pseudo-targeted dataset. Hence, compound **34** was tentatively recognized as romerine.

Compounds **27** and **37** have the same protonated ions at *m*/*z* 312.1594, corresponding to the molecular formula of C_19_H_21_NO_3_, which has one more oxygen atom than nuciferine. The fragment ion at *m*/*z* 281.1158 of compound **27** was produced by a CH_3_NH_2_ loss from the protonated ion, implying the oxygen atom was not at N-2. Compound **37** has the characteristic fragment ion at *m*/*z* 265.1216 in the MS^2^ spectrum, which indicates that the position of N-2 was replaced by a hydroxymethyl group rather than a methyl group. According to the pseudo-targeted dataset, they were tentatively assigned as oxidation-nuciferine and nuciferine-*N*-methanol, respectively. Similarly, Compound **30** was tentatively recognized as *N*-methyl asimilobine *N*-oxide, which has a protonated ion at *m*/*z* 298.1438. Compound **41** has a protonated ion at *m*/*z* 324.1594, which provided fragment ions at *m*/*z* 265.1228, 250.1053, 233.0961 and 218.0728 in the MS^2^ spectrum. The neutral loss of acetamide (−59.0366 Da) indicates that the N-2 position was replaced by an acetyl group. Thus, compound **41** was tentatively identified as nuciferine-*N*-acetyl. Compound **42** has a protonated ion at *m*/*z* 308.1281, displaying a series of fragment ions at *m*/*z* 249.0906, 219.0802, 191.0850 and 165.0690 in the MS^2^ spectrum, which is consistent with those of anonaine-*N*-acetyl in the pseudo-targeted dataset. Thus, compound **42** was tentatively regarded as anonaine-*N*-acetyl.

Compounds **9** and **18** have protonated ions at *m*/*z* 298.1438 and 312.1594, respectively. Their MS^2^ spectral features are similar to those of glaziovine and pronuciferine, respectively, in the developed dataset. Thus, compounds **9** and **18** were tentatively identified as glaziovine and pronuciferine, respectively. 

#### 3.2.4. Identification of Amide Alkaloids

In the case of amide alkaloids (Figure 5D), the characteristic fragment ions of these alkaloids are produced by the cleavage of amide bonds. 

Compound **38** was eluted from an LC column at 6.18 min with a protonated ion ([M + H]^+^) at *m*/*z* 314.1401 (Figure 9). Diagnostic fragment ions at *m*/*z* 177.0540 and 121.0633 were observed in the MS^2^ spectrum, which originated from the cleavage of the amide bond. In addition, the fragment ion at *m*/*z* 145.0272 was produced by the neutral loss of a CH_3_OH from the fragment ion at *m*/*z* 177.0540. By comparison of the retention time and the mass spectral data with those of alkaloid standard, compound **38** was verified as trans-*N*-feruloyltyramine. Similarly, Compound **1** and compound **36**, with the protonated ions at *m*/*z* 123.0552 and *m*/*z* 314.1395, were verified as nicotinamide and cis-*N*-feruloyltyramine, respectively. Compounds **39** and **40** have the same protonated ions at *m*/*z* 268.1332 and fragment ions at *m*/*z* 131.0494, 121.0653 and 103.0548 in their MS^2^ spectra, which is consistent with the structural characteristics of *N*-cinnamoyltyramine in the pseudo-targeted dataset. Thus, compounds **39** and **40** were tentatively assigned as the cis/trans isomers of *N*-cinnamoyltyramine.

Finally, a total of 42 alkaloids were tentatively annotated from the ethyl lactate extracts of lotus seed epicarp. Except nuciferine, isoliensinie and neferine, the remaining 39 alkaloids were tentatively annotated in lotus seed epicarp for the first time.

### 3.3. Analysis of Ten Representative Alkaloids in Lotus Seed Epicarp at Different Growth Stages

Ten representative alkaloids in lotus seed epicarp were quantified by UPLC-QTOF-MS/MS with high-resolution multiple-reaction monitoring (MRM^HR^) mode. A T3 column and a mobile phase system containing 0.1% formic acid in water (A) and 0.1% formic acid in acetonitrile (B) were selected to obtain an excellent separation effect and ionization effect.

#### 3.3.1. Validation of Quantitative Analysis

The proposed quantitative method was validated by the linear range, correlation coefficient (R^2^), limits of detection (LODs), limits of quantification (LOQs), precision and accuracy. As shown in Appendix A, ten alkaloids showed good linearity within the test concentration ranges (R^2^ ≥ 0.996). LODs and LOQs of ten alkaloids were in the range of 0.10–0.52 ng/mL and 0.50–3.00 ng/mL, respectively. The RSD values of the intraday and interday variations were in the ranges of 0.68–4.79% and 2.00–5.26%, respectively. The recoveries at three spiked levels were in the range of 81.09–102.03%, and the RSD value was in the range of 0.80–8.75% (Appendix A). In summary, the developed method is credible for the determination of ten alkaloids in lotus seed epicarp.

#### 3.3.2. Determination of Ten Alkaloids in Lotus Seed Epicarp at Different Growth Stages

According to the growing periods, lotus seed epicarp was classified into five stages (I, II, III, IV and V). As shown in Figure 10, the color of lotus seed epicarp changed from light yellow (stage I) to yellowish green (stage III) and then to black–brown (stage V). The total amount of the ten alkaloids changed with growing periods, which increased gradually from stage I to III, reaching its maximum at stage III. Among the ten studied alkaloids in the lotus seed epicarp, N-nornuciferine, nuciferine and N-methycoclaurine are the predominant alkaloids in the five growth stages. The content of N-methycoclaurine was the highest (2034.13 ± 2.83 µg/g) in stage II, whereas N-nornuciferine content was the highest (4555.33 ± 26.40 µg/g) in stage III, and nuciferine was the highest (2534.67 ± 20.27 µg/g) in stage IV (Appendix A). Interestingly, the distributions and trends of alkaloids in lotus seed epicarp are consistent with those in lotus leaves, indicating that lotus seed epicarp may have comparable healthcare function to that of lotus leaves [47,48,49].

#### 3.3.3. Principal Component Analysis (PCA)

Distribution of the ten alkaloids in lotus seed epicarp in different growth stages were evaluated using PCA analysis. The score plot of the first two principle components is shown in Figure 11; the lotus seed epicarp at different growth stages could be well differentiated by PC1 (41.1%) and PC2 (30.2%). This result suggests that the ten alkaloids can be used to distinguish the lotus seed epicarp at different growth stages. 

## 4. Conclusions

In this study, an ultrasonic-assisted extraction method with ethyl lactate as the extraction agent was developed to extract alkaloids from lotus seed epicarp. The extraction conditions were investigated by the response surface methodology. Compared to previously reported extraction methods, the proposed method for extraction of alkaloids from lotus seed epicarp is environmentally friendly and efficient, with less solvent consumed and shorter ultrasonic time. Interestingly, the ethyl lactate extracts of mature lotus seed epicarp (stage V) have the highest DPPH scavenging rate. Moreover, applying the pseudo-targeted method, a total of 42 alkaloids were tentatively identified from the lotus seed epicarp extracts by UPLC-QTOF-MS/MS with IDA. According quantitative analysis, the distributions and trends of alkaloids in the lotus seed epicarp were similar to those of lotus leaves. In conclusion, ethyl lactate could be used as an alternative to traditional solvent to extract alkaloids from lotus seed epicarp, and lotus seed epicarp could be a potential renewable resource for medicinal applications and as functional food. 

## Figures and Tables

**Figure 1 foods-11-01056-f001:**
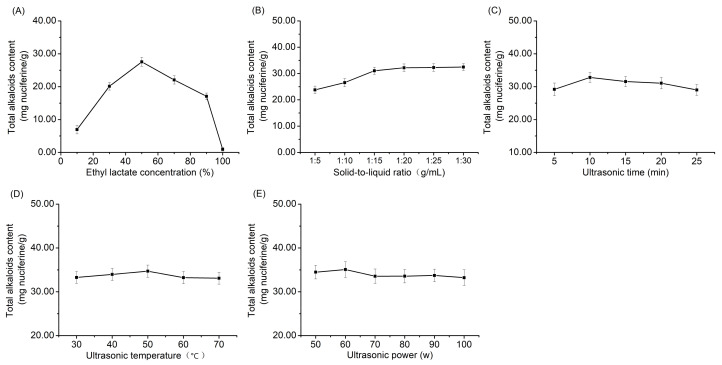
Effects of (**A**) ethyl lactate concentration, (**B**) solid-to-liquid ratio, (**C**) ultrasonic time, (**D**) ultrasonic temperature and (**E**) ultrasonic power on the total alkaloid content extracted from lotus seed epicarp.

**Figure 2 foods-11-01056-f002:**
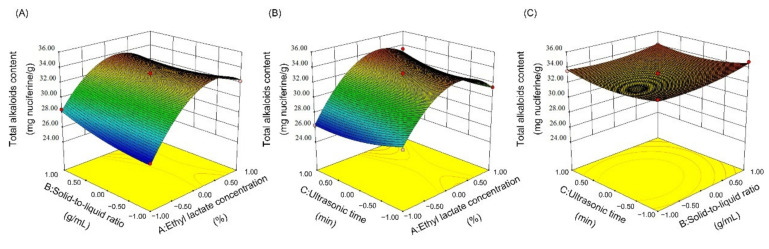
3D response surface of the total alkaloid content affected by the interaction between (**A**) ethyl lactate concentration and solid-to-liquid ratio, (**B**) ethyl lactate concentration and ultrasonic time and (**C**) solid-to-liquid ratio and ultrasonic time.

**Figure 3 foods-11-01056-f003:**
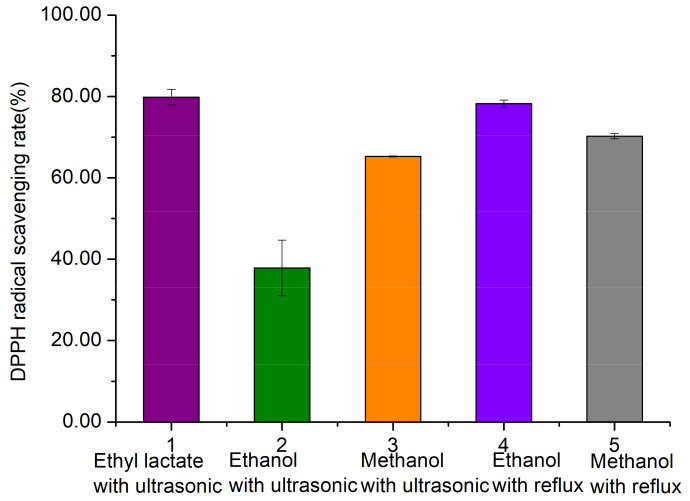
The DPPH radical scavenging abilities of lotus seed epicarp extracts obtained by different extraction methods.

**Figure 4 foods-11-01056-f004:**
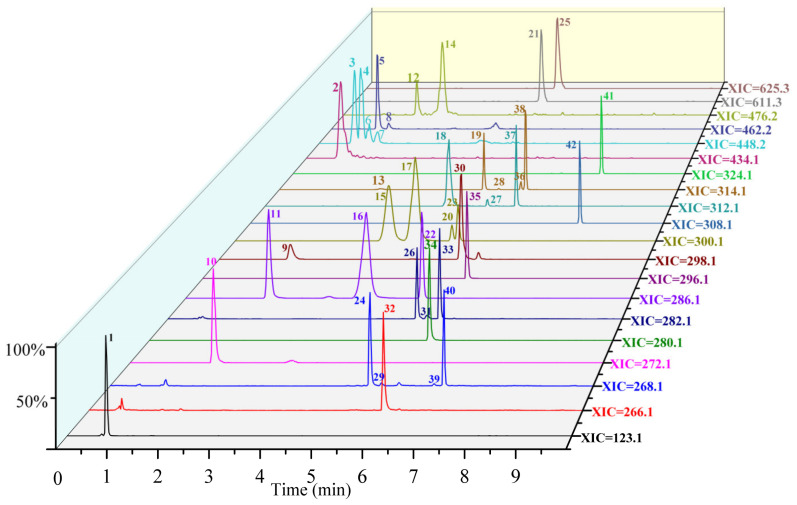
The extracted ion chromatograms of 42 alkaloids in lotus seed epicarp.

**Figure 5 foods-11-01056-f005:**
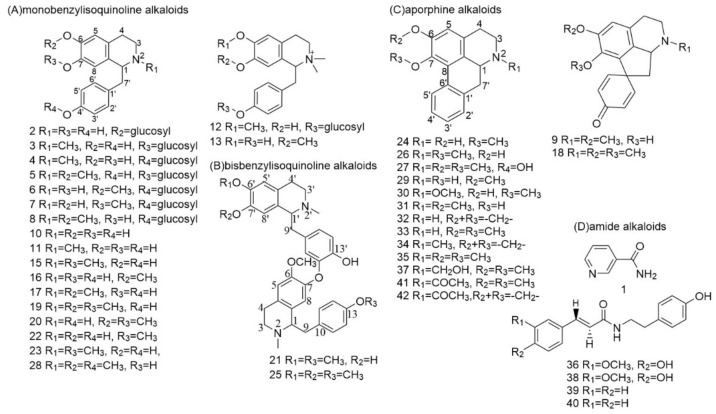
Structure of 42 alkaloids from lotus seed epicarp.

**Figure 6 foods-11-01056-f006:**
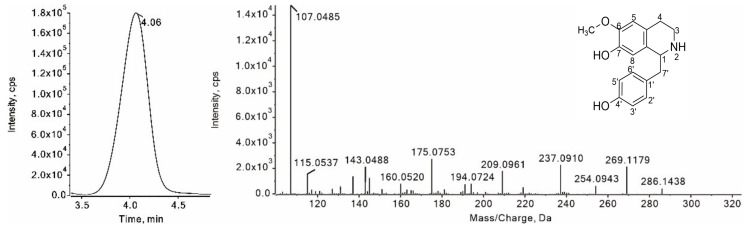
XIC and the MS/MS spectrum of coclaurine.

**Figure 7 foods-11-01056-f007:**
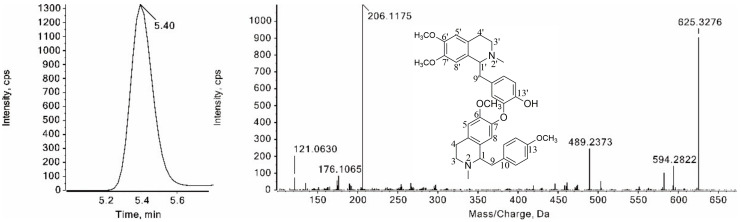
XIC and the MS/MS spectrum of neferine.

**Figure 8 foods-11-01056-f008:**
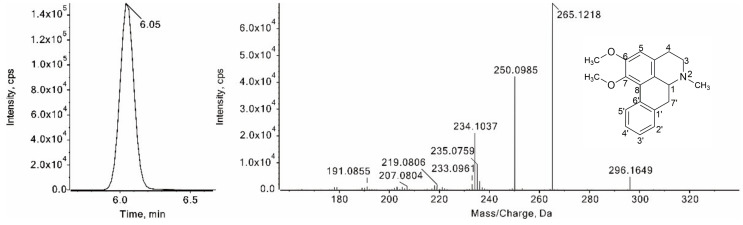
XIC and the MS/MS spectrum of nuciferine.

**Figure 9 foods-11-01056-f009:**
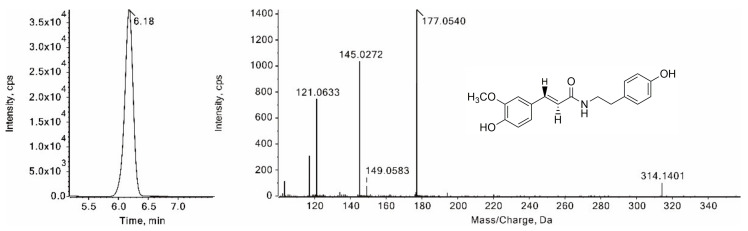
XIC and the MS/MS spectrum of trans-*N*-feruloyltyramine.

**Figure 10 foods-11-01056-f010:**
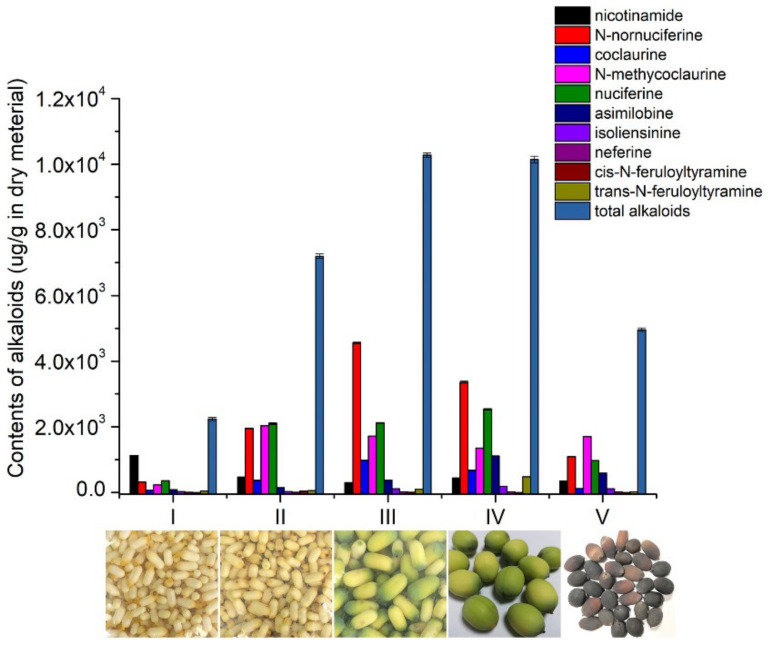
Contents of ten alkaloids in lotus seed epicarp at different growth stages.

**Figure 11 foods-11-01056-f011:**
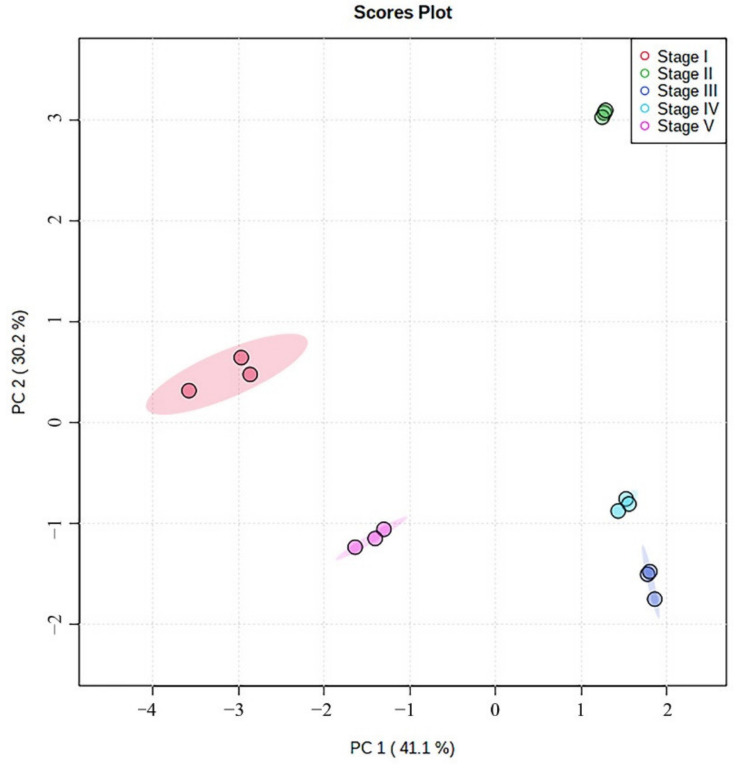
Score plot for five different growth stages of lotus seed epicarp.

**Table 1 foods-11-01056-t001:** The total alkaloid content extracted from lotus seed epicarp by different extraction methods.

Method	Solid-to-Liquid (g/mL)	Extraction Time (min)	Temperature (°C)	Total Alkaloid Content (mg Nuciferine/g)	Ref.
Ethyl lactate with ultrasonic	1:10	15	50	34.75 ± 1.55	This work
Ethanol with ultrasonic	1:25	30		27.484 ± 1.56	[34]
Methanol with ultrasonic	1:50	30		35.55 ± 1.42	[35]
Ethanol with reflux	1:10	60	90	35.69 ± 1.51	[11]
Methanol with reflux	1:10	60	25	21.29 ± 1.53	[36]

## Data Availability

The data presented in this study are available in Appendix A.

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
