# Peer review of "Green Extraction-Assisted Pseudo-Targeted Profile of Alkaloids in Lotus Seed Epicarp Based on UPLC-QTOF MS with IDA"

_foods, 2022, doi:10.3390/foods11071056_

Round 1

Reviewer 1 Report

The document “Green extraction assisted pseudo-targeted profile of alkaloids in lotus seed epicarp based on UPLC-QTOF MS with IDA” is an interesting manuscript well-presented, performed and discussed that focus on the instrumental analysis determination that has been carried out in a very interesting way, some comments need to be addressed.

In introduction section information of ultrasonic extraction could be added, as an example de document DOI: 10.3390/agronomy7030047 contains general information. In this sense, more information about the solvent and other similar works must be added in this section.

In materials and methods section, information about the refence of the sonication conditions must be added.

In results and discussion section authors indicates that the high viscosity of the extract solution is unfavorable for the mass transfer, nevertheless the viscosity parameter is not measured.

Why did the authors use peak area intensity of total alkaloids instead of a value such as “equivalent of X compounds” or the identified compounds?

The quality of figures must be improved specially figures from 5 to 8.

Author Response

Dear reviewer,

Thank you for your careful review of our manuscript entitled “Green extraction assisted pseudo-targeted profile of alkaloids in lotus seed epicarp based on UPLC-QTOF MS with IDA” (foods-1645896) which has been submitted to foods. We have revised our manuscript according to the comments of the reviews one by one. The details are as follows:

1) In introduction section information of ultrasonic extraction could be added, as an example de document DOI: 10.3390/agronomy7030047 contains general information. In this sense, more information about the solvent and other similar works must be added in this section.

Response: Thanks for your advice, the information about the ultrasonic extraction, the solvent and the other similar works have been added in the introduction section of the revised manuscript.

2) In materials and methods section, information about the refence of the sonication conditions must be added.

Response: The refence of the sonication conditions have been added as “2.5 Conventional reference extraction methods” in the revised manuscript and detailed as “S3 Conventional reference extraction methods” in the supplementary materials.

3) In results and discussion section authors indicates that the high viscosity of the extract solution is unfavorable for the mass transfer, nevertheless the viscosity parameter is not measured.

Response: The viscosity of pure water and pure ethyl lactate are 0.89 and 2.21 mPa.s at 25 ℃ under atmospheric pressure,respectively. Santiago Aparicio et al. (doi:10.1021/jp904668e.) have systematacially investigated the viscosity of water and ethyl lactate mixture, which would be effected by pressure, temperature and mixture composition. In addition, the dynamic viscosity of mixture solvent correlates with mole fraction in a nonlinear mode. When the mole fraction of ethyl lactate was at 40% (approximately equal to the volume fraction of 81.4%(v/v)), the dynamic viscosity of mixture solvent reached the maxima. The phenomenon described in our manusript could be well explanined by the rule reported in the literature. Due to the limited experiment equipment, the determination of the viscosity of ethyl lactate aqueous solution has not been considered in our current work.

4) Why did the authors use peak area intensity of total alkaloids instead of a value such as “equivalent of X compounds” or the identified compounds?

Response: The “peak area intensity of total alkaloids” has been displaced with “equivalent of nuciferine” in the revised manuscript.

 5) The quality of figures must be improved specially figures from 5 to 8.

Response: Thank you very much for your advice. All figures have been improved as shown in the revised manuscript.

Thank you very much for your help.

Sincerely yours,

Xiaoji Cao

Reviewer 2 Report

Dear authors, 

I reviewed your manuscript and found it interesting. However, the material and methods must be rewritten because so many details were forgotten. Below, you will see details of my review. 

Introduction

It should be more information about agrohain of lotus in order to give support to work.

How Ethyl lactate can be classified? NADES? Added more details about your extraction solvent because it is a novelty.

Also, the authors should add more detail about sonication in the introduction.  Advantages and Disadvantages.

Material and methods

What was the sonication device used in extraction? 60W is very low power. The optimization study was forgotten in this section. Very details must be given here. How were the conditions selected? 

What was the experimental design employed?

It is not clear as the conditions were combined. Clarify, please!

Statistical analysis item is forgotten too. The information given on results and discussion must be moved for material and methods. 

Results

How was the extraction efficiency calculated? It must be described in material and methods.

3.3 Analysis of ten representative alkaloids in lotus seed epicarp at different growth stages .... It is not described in the material and methods section.

PCA analysis (MetaboAnalyst). It must be described in statistical analysis.

The antioxidant capacity of the extracts must be evaluated as a minimal response to the study. In vitro biological assays should be performed in order to improve work quality. 

Author Response

Dear reviewer,

Thank you for your careful review of our manuscript entitled “Green extraction assisted pseudo-targeted profile of alkaloids in lotus seed epicarp based on UPLC-QTOF MS with IDA” (foods-1645896) which has been submitted to foods. We have revised our manuscript according to the comments of the reviews one by one. The details are as follows:

1) It should be more information about agrohain of lotus in order to give support to work.

Response: Thank you very much for your advice, the information about agrohain of lotus has been added in the introduction section of the revised manuscript.
2)How Ethyl lactate can be classified? NADES? Added more details about your extraction solvent because it is a novelty.

Response: Ethyl lactate is classified as bio-based solvents. And the information about the ethyl lactate has been detailed in the introduction section of the revised manuscript.
3) Also, the authors should add more detail about sonication in the introduction.  Advantages and Disadvantages.

Response: The information about sonication has been added in the introduction section of the revised manuscript.
4) What was the sonication device used in extraction? 60W is very low power. The optimization study was forgotten in this section. Very details must be given here. How were the conditions selected?

Response: A KQ-100DB ultrasonic cleaning bath (Shumei, Kunshan, Jiangsu, China) was used in extraction. The optimization study and the conditions selection has been added as “2.4 Optimization of ultrasonic extraction method” in the revised manuscript.

The single-factor experiments were detailed in supplementary materials.
5) What was the experimental design employed?

Response: Single-factor experiments and the response surface methodology experiment with a three-factors-three-levels Box-Behnken design were employed in the revised manuscript.

6) It is not clear as the conditions were combined. Clarify, please!

Response: The conditions have been clarified as “2.7 Conditions of UPLC-QTOF-MS/MS” in the revised manuscript.

7) Statistical analysis item is forgotten too. The information given on results and discussion must be moved for material and methods.

Response: The statistical analysis item has been added as “2.9 Statistical analysis” in the revised manuscript. The information given on results and discussion has been moved to material and methods.

8) How was the extraction efficiency calculated? It must be described in material and methods.

Response: In the revised manuscript, the extraction efficiency was evaluated by the total alkaloids content, which was expressed as mg of nuciferine per gram of dried lotus seed epicarp (mg nuciferine/g). The detail was shown in supplementary materials.

9) 3.3 Analysis of ten representative alkaloids in lotus seed epicarp at different growth stages .... It is not described in the material and methods section.

Response: The experiment of “3.3 Analysis of ten representative alkaloids in lotus seed epicarp at different growth stages” has been added as “2.8 Quantification of ten representative alkaloids in lotus seed epicarp at different stages” in the material and methods section of the revised manuscript.

10) The antioxidant capacity of the extracts must be evaluated as a minimal response to the study. In vitro biological assays should be performed in order to improve work quality.

Response: The DPPH radical scavenging rate of the lotus seed epicarp extracts by different extraction method has been evaluated in the revised manuscript.

Thank you very much for your help.

Sincerely yours,

Xiaoji Cao

Round 2

Reviewer 2 Report

Dear Author, 

Congratulations on your work after review.